# Buffer Overflow in Mixture of Experts

**Jamie Hayes**
Google DeepMind

**Ilia Shumailov**
Google DeepMind

**Itay Yona**
Google DeepMind

## Abstract

Mixture of Experts (MoE) has become a key ingredient for scaling large foundation models while keeping inference costs steady. We show that expert routing strategies that have cross-batch dependencies are vulnerable to attacks. Malicious queries can be sent to a model and can affect a model's output on other benign queries if they are grouped in the same batch. We demonstrate this via a *proof-of-concept* attack in a *toy experimental setting*.

## 1 Overview

In the early days of computing, time-sharing gave a major boost to user productivity and hardware utilisation. Yet, it also brought significant security challenges; computers were now shared, meaning unauthorised access had to be mediated, while individual private data had to be protected through explicit isolation. Machine Learning (ML) is now experiencing its time-sharing moment – hardware costs and high utilisation requirements are pushing accelerators that host ML models to be shared across different users, naturally leading to a question: *are there any new security concerns*?

In this work we demonstrate that when Mixture of Experts (MoE) is used in, for example, a transformer architecture (Vaswani et al., 2017), an adversary can change the model prediction on data of other users who happen to be placed into the same batch. This is because many routing strategies that assign inputs to experts are not inter-batch data (order) invariant. This vulnerability affects current MoE implementations that set a buffer capacity limit on the number of inputs each expert can process.

We describe two ways to generate adversarial data and show that both provide an adversary with an ability to run integrity attacks  cause deterministic fault injection, as well as, availability attacks denial of expert attacks against other users. Finally, we investigate several mitigation strategies that nullify the attack or significantly reduce its efficiency.

## 2 Technical details

We first describe MoE (Jacobs et al., 1991; Jordan and Jacobs, 1994) and its later sparse variants (Du et al., 2022; Fedus et al., 2022; Riquelme et al., 2021; Shazeer et al., 2017), and following this, introduce attack details.

### 2.1 Mixture of Experts

We follow the set-up described in Riquelme et al. (2021) for the deep learning setting. Given an input token $z \in R^d$, the output of $z$ through an MoE layer is given by:

$$MoE(z) = \sum_{i=1}^{n} g_i(z)e_i(z), \tag{1}$$

Neurips Safe Generative AI Workshop 2024.

where $e_i : R^d \rightarrow R^d$ are the $n$ experts, and $g_i : R^d \rightarrow R$ is a gating function that assigns an input conditioned weight to an expert. In the settings we consider, $g$ and $e$ are parameterized by fully-connected neural networks. MoE is referred to as *sparse* if $g$ assigns non-zero weight to fewer than $n$ experts. This is can save on inference costs as the outputs of all $n$ experts do not need to be evaluated. In most MoE implementations, the number of zero weights is preset by only routing the top-$k$ values of $g$ to their assigned experts, where $k < n$. After a subset of top-$k$ values of $g$ have been selected, it is common to adjust the weights in eq. (1) by re-normalizing this subset through a softmax function.

One of the primary motivations for using MoE is improved model performance for a fixed number of activated parameters (Shazeer et al., 2017). With MoE, the whole network is not needed to evaluate a given input. For example, the popular open-source MoE transformer model, Mixtral-8×7B (Jiang et al., 2024), has 46.7B parameters, but effectively only uses a smaller subset of 12.9B parameters per token.

## 2.2 Routing strategies

Our investigation focuses on transformer models that use MoE, where inputs to the model are sequences of tokens from a vocabulary. Let $Z \in R^{B \cdot T \times d}$ be an input to the MoE layer, where $B$ is the batch size, $T$ is the number of tokens per sequence in the batch, and $d$ is a $d$-dimensional representation of a token at this layer. For each $z \in Z$, we first compute $g(z) = \{g_1(z), g_2(z), \ldots, g_n(z)\}$, where $n$ is the number of experts. This gives the matrix

$$G = \begin{bmatrix} g_{11} & g_{12} & \cdots & g_{1n} \\ g_{21} & g_{22} & \cdots & g_{2n} \\ \vdots & \vdots & \vdots & \vdots \\ g_{(B \cdot T)1} & g_{(B \cdot T)2} & \cdots & g_{(B \cdot T)n} \end{bmatrix}, \tag{2}$$

where $g_{ij}$ denotes the routing weight of token $z_i$ to expert $e_j$. There are various methods to rank and prioritise how tokens are sent to experts. Fedus et al. (2022); Lepikhin et al. (2020); Riquelme et al. (2021); Shazeer et al. (2017) use variations of the following strategy, which is referred to as the *vanilla* routing strategy: Sequentially, starting from the first row of $G$, find the column with top-1 value for that row and send that token to the expert associated with that column. Once top-1 assignments have been made for all $B \cdot T$ tokens, repeat for top-2 assignments, and keep repeating this until all top-$k$ assignments for each row have been made. This routing strategy is given in pseudo-code in Algorithm 1.

---
**Algorithm 1** Vanilla expert routing strategy
---

**Args:** number of experts $n$, number of experts per token $k$, batch size $B$, number of tokens per example $T$, dimensionality of token representation $d$, expert functions $e_i : R^d \rightarrow R^d$ ($i \in \{1, \ldots, n\}$), gating function $g : R^d \rightarrow R^n$, input to MoE layer $Z \in R^{B \cdot T \times d}$.
$G \leftarrow$ Initialize a $(B \cdot T) \times n$ matrix by passing each $z \in Z$ into the gating function $g$.
$h = 1$
**while** $h \leq k$ **do**
    $i = 1$
    **while** $i \leq B \cdot T$ **do**
        $g_{ij} \leftarrow$ Get $jth$ column for row $i$ of $G$ (the column with the top-$h$ value).
        Assign $z_i$ to expert $e_j$
        $i = i + 1$
    $h = h + 1$

---

In theory, each expert in Algorithm 1 can process $B \cdot T$ tokens. This could happen if, for each $k$, the top-$k$ value in each row of $G$ had the same expert assignment. An uneven assignment of experts is generally an undesirable property as this results in under utilization of some experts. To partially mitigate this issue, a buffer capacity limit is usually set for each expert (Gale et al., 2023; Riquelme et al., 2021). This is a static number, $B_e$, that dictates how many tokens each expert can process, which in practice is set by a capacity slack variable $C$, where:

$$B_e = \text{round}\left(\frac{kCBT}{n}\right). \tag{3}$$

If $C < 1$, some routing assignments will be ignored, while imbalances in routing assignments between experts can be tolerated if $C > 1$. We refer the interested reader to Fedus et al. (2022), who discuss the trade-offs in setting the buffer capacity limit.

## 2.3 Threat model and attack method

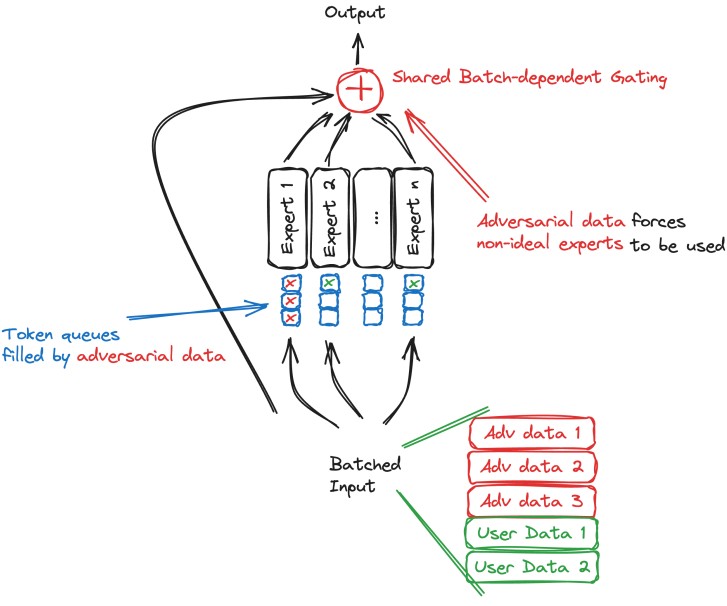

Figure 1: *Overall attack flow. The adversary pushes their data into the shared batch, that already contains user data. As tokens get distributed across different experts, adversarial data fills the expert buffers that would be preferred by the user, dropping or routing their data to experts that produce suboptimal outputs.*

Our attack relies on MoE routing that uses finite sized buffer queues for each individual expert. When these queues are filled, tokens could be dropped [1] or passed to another expert whose queue is not full (Fedus et al., 2022). This opens up a new attack vector; if adversarial data is mixed with benign user data in a batch, the adversarial data can influence the expert choice of the benign data by filling the buffer of certain experts. We depict the overall attack process in Figure 1. For the vanilla routing strategy in Algorithm 1, tokens are sequentially assigned to experts according to the order they appear in the batch. This means inputs at the beginning of the batch can affect the routing assignments of tokens further down in the order. Imagine there are two data-points in the batch $x_1$ and $x_2$, each point has two tokens, $\{z_i^1, z_i^2\}$, and there are two experts $e_1$ and $e_2$ with a buffer capacity of two. The input to the MoE layer is therefore $\{z_1^1, z_1^2, z_2^1, z_2^2\}$, and let's assume that all tokens should be sent to $e_1$. According to the vanilla routing strategy, the first experts buffer will be filled first with $z_1^1$ and then $z_1^2$, and so the assignments for $z_2^1$ and $z_2^2$ are ignored. Crucially, because each expert has a buffer capacity limit, the assignment of tokens to experts is batch dependent — one data-point can change the expert assignment of another data-point. Our attack exploits this inter-batch routing dependency to find adversarial data-points that can negatively affect the outputs of other data-points in the batch.

Our attack set up is as follows: Let $f_\theta : \mathcal{V}^T \to R^s$ be a transformer model that uses MoE layers and is parameterized by a set of weights $\theta$, where $\mathcal{V}^T$ denotes a set of $T$ tokens from a vocabulary

---
[1]Note, although dropping tokens is generally considered to be wasteful, they still influence downstream layers as they are directly passed to the next layer through a residual connection. Our experiments will drop tokens when an expert buffer is full rather than re-route to different experts.

$\mathcal{V}$ of size $s$. The transformer model takes as input a sequence of $T$ tokens and outputs a probability distribution over $\mathcal{V}$. It is typical to batch inputs together to maximize throughput, we assume that given a batch of inputs $X = \{x_1, x_2, \ldots, x_{B-1}, x^*\}$, where each $x \in X$ is a set of $T$ tokens, that the adversary controls all elements in the batch other than $x^*$. We refer to this subset of adversarially controlled inputs as $X^{\mathcal{A}}$. Note that $f_\theta(X) \in R^{B \times s}$, and we denote the output of $x^*$ as $f_{\theta,x^*}(X = X^{\mathcal{A}} \cup \{x^*\}) \in R^s$. We also assume the adversary can control the order of batch elements, and always ensures that $x^*$ is last in the batch. Let's assume that the next predicted token in the sequence after $x^*$ should be $y \in \mathcal{V}$ [2] and before the attack, $\arg\max f_{\theta,x^*}(X) = y$. The attack goal is to find adversarial inputs $X^{\mathcal{A}}$ such that $\arg\max f_{\theta,x^*}(X) \neq y$. We find this by optimizing $X^{\mathcal{A}}$ to minimize the loss $l_{X^{\mathcal{A}}}(x^*) = f_{\theta,x^*}(X^{\mathcal{A}} \cup \{x^*\})_y - \arg\max_{\hat{y} \neq y} f_{\theta,x^*}(X^{\mathcal{A}} \cup \{x^*\})$, where $f_{\theta,x^*}(X^{\mathcal{A}} \cup \{x^*\})_y$ denotes the output at token $y$ in $\mathcal{V}$. We refer to the set of adversarial inputs after this optimization process as $\tilde{X}^{\mathcal{A}}$.

Throughout our experiments we use simple random search to construct $\tilde{X}^{\mathcal{A}}$. We randomly sample a subset of tokens from the vocabulary and check if this decreases $l_{X^{\mathcal{A}}}(x^*)$. The step-by-step process of this search is given in Algorithm 2.

---

**Algorithm 2** Random search attack

---

**Args:** Loss function $l$, batch size $B$, number of tokens per sample in the batch $T$, number of iterations $M$, number of tokens to change per iteration $r$, input $x^*$ with target $y$, vocabulary $\mathcal{V}$.
**Output:** Optimized adversarial data $\tilde{X}^{\mathcal{A}}$
$X^{\mathcal{A}} \leftarrow \{z_1^1, z_1^2, \ldots, z_1^T, z_2^1, \ldots, z_{B-1}^T\}$ ▷ Initialize $(B-1) \cdot T$ tokens at random from $\mathcal{V}$.
smallest loss $\leftarrow 999$ ▷ We track the adversarial inputs that achieve the smallest loss on $l(\cdot)$.
$\tilde{X}^{\mathcal{A}} \leftarrow X^{\mathcal{A}}$ ▷ Initialize the current best choice of adversarial inputs to $\tilde{X}^{\mathcal{A}}$.
$i = 1$
**while** $i \leq M$ **do**
    loss $\leftarrow l_{X^{\mathcal{A}}}(x^*)$ ▷ Compute the loss on $x^*$.
    **if** loss $<$ smallest loss **then** ▷ Check if current loss is smaller than any previously computed loss.
        smallest loss $\leftarrow$ loss ▷ Assign the tracked smallest loss to current loss.
        $\tilde{X}^{\mathcal{A}} \leftarrow X^{\mathcal{A}}$ ▷ Assign the best choice of adversarial inputs to the current ones.
    **else**
        $X^{\mathcal{A}} \leftarrow \tilde{X}^{\mathcal{A}}$ ▷ Re-initialize the current adversarial inputs to the tracked adversarial inputs that achieved the smallest loss so far.
    $X^{\mathcal{A}} \leftarrow$ Replace $r$ tokens at random.
    $i = i + 1$
**return** $\tilde{X}^{\mathcal{A}}$ ▷ Return the optimized adversarial inputs.

---

This attack assumes that an adversary is interacting with a black-box model that is utilising MoE, and observes logit outputs. It also assumes the adversary can ensure their data is always grouped in the same batch as the target point $x^*$. We make no assumptions about the MoE configuration, except that it has per-expert queues that are finite and uses the batch order dependent vanilla routing strategy. This final assumption is not unreasonable and is how many MoE based transformer models are trained and deployed.

## 3 Attack demonstration

We demonstrate our attack on the popular open-source MoE transformer model, Mixtral-8×7B, which has a vocabulary size of 32,000, uses 8 experts (per layer), and each token can be assigned to a maximum of 2 experts. We set the target $x^*$ to the prompt "`Solve the following equation: 1+1=`" which has a sequence length of 12 tokens. We confirmed that the most likely next token output by the model is "2", with a softmax probability of 25.9% [3]. We set the batch size to 8, and so there are a total of $(B = 8) \times (T = 12)$ tokens in the batch. Mixtral-8×7B does not set an expert buffer capacity limit in its default MoE implementation and so is **_not_** vulnerable to our attack. We augment the MoE to use the vanilla routing strategy with an expert buffer capacity of 38 tokens (out

---

[2] We abuse notation and let $y$ represent both the vocabulary element *and* index within $\mathcal{V}$.
[3] One may wonder why this probability is so low. This is likely because we are using the base model that has not been instruction tuned and we do not use few-shot learning.

of a possible 96). We confirmed that this change did not affect the model's output on $x^*$. However, such a small buffer capacity will almost certainly affect the quality of outputs on longer sequences.

The number of attack iterations is set to 1,000, and we modify 5 tokens per data-point in the adversarial controlled batch $X^{\mathcal{A}}$ per iteration. The number of tokens per data-point is 12 and there are 7 data-points in $X^{\mathcal{A}}$, so the adversary can modify a total of 35 tokens out of a possible 84 per iteration.

Results of the attack are shown in Figure 2. Initially, "2" has the largest probability, and the next most likely token "1" has a probability $\approx 20\%$. At the beginning of the attack, the number of tokens assigned to the second expert from the adversarial batch $X^{\mathcal{A}}$ is 37 and there is one token assigned to this expert from $x^*$. At iteration 60, the buffer of the second expert is filled entirely with tokens from $X^{\mathcal{A}}$, which means the token in $x^*$ that was assigned to this expert is dropped. This change in expert assignments (and expert assignments in downstream layers) is enough to cause "1" to become the most likely next token. We also plot the difference in softmax probabilities between tokens "2" and "1" throughout the attack.

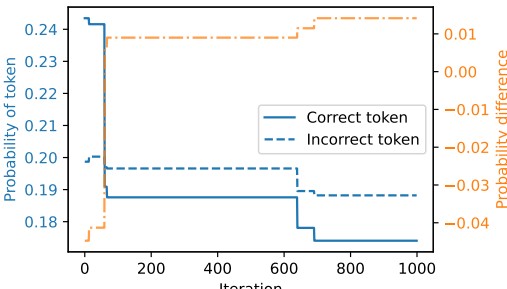

Figure 2: *Probability of correct token, "2", and (most likely) incorrect token, "1", throughout the random search attack. By the end of the attack the output token with largest probability is the incorrect token, "1".*

# 4 Anecdotal evidence of transferability to different prompts

It is unrealistic to assume an adversary ensure that $\tilde{X}^{\mathcal{A}}$ will always grouped with a target point. The attack becomes much more practical if one can show that errors can be induced on data-points other than $x^*$ when included in a batch with $\tilde{X}^{\mathcal{A}}$. This opens up the possibility of untargeted attacks, where an adversary constructs $\tilde{X}^{\mathcal{A}}$ that block prior chosen experts that are preferred for certain tasks (e.g. arithmetic or translation).

We check how well the optimized adversarial data-points, $\tilde{X}^{\mathcal{A}}$, transfer to arithmetic prompts other than $x^*$. We conjecture that if these other data-points are sampled from the same distribution as $x^*$, they are more likely to have similar expert routing assignments, and so are more likely to be affected by $\tilde{X}^{\mathcal{A}}$, which was optimized specifically for $x^*$. We measure the change in most likely next token probabilities on prompts that are similar to $x^*$ in the Table 1.

Table 1: *Transferability of $\tilde{X}^{\mathcal{A}}$ constructed against $x^*$ ("Solve the following equation: 1+1=").*

| Prompt & completion before batching with $\tilde{X}^{\mathcal{A}}$ | Prompt & completion after batching with $\tilde{X}^{\mathcal{A}}$ | Probability change of the correct token | Did the attack induce an error? |
|---|---|---|---|
| "Solve the following equation: 2+2=4" | "Solve the following equation: 2+2=4" | $-1.5\%$ | No |
| "Solve the following equation: 4+4=8" | "Solve the following equation: 4+4=4" | $-1.8\%$ | Yes |

The attack successfully transferred to one other prompt; the inclusion of the adversarial data-points in the batch induced an error in the model's output on this prompt. Even for the prompt where the attack failed to induce an error, the probability of the correct token, "4", was reduced. This is anecdotal evidence that adversarial data can transfer and induce model output errors on prompts unseen during the construction of $\tilde{X}^{\mathcal{A}}$.

# 5   Does the position in batch matter?

Any expert routing strategy that uses an expert buffer capacity limit that is smaller than the total number of tokens in the batch will be vulnerable to an attack that attempts to fill up certain expert buffers to block other token's assigned experts. The most vulnerable data-point in the batch depends on the MoE routing strategy. The vanilla routing strategy sequentially assigns experts by position in the batch, and so a data-point that is last in the batch is more vulnerable than data-points that are near the front, as there are more opportunities for other data-points to change the expert assignment of the tokens from the targeted data-point. We verified this by running the random search attack while always placing $x^*$ to be first in the batch; the attack did not induce an error in the model's output on $x^*$. However, we note that the expert assignments of a data-point that is first in the batch can still be affected by other points if the vanilla routing strategy assigns tokens to multiple experts, i.e., uses top-$k$ gating values where $k > 1$. Consider the previous illustrative example where four tokens $\{z_1^1, z_1^2, z_2^1, z_2^2\}$ can be sent to experts $e_1$ or $e_2$, each with a buffer capacity of two. If we set the targeted point to $x^* = x_1 = \{z_1^1, z_1^2\}$, and let the adversary control $\tilde{X}^{\mathcal{A}} = x_2 = \{z_2^1, z_2^2\}$, then $\tilde{X}^{\mathcal{A}}$ cannot affect the top-1 expert assignments of $x^*$, but can change the top-2 assignment. For example, if the top-2 assignment for $z_1^1$ is $e_2$ but the top-1 assignments for $\tilde{X}^{\mathcal{A}}$ are also $e_2$, then $z_1^1$ would not be assigned to $e_2$ as its buffer has been filled with the adversarial tokens.

# 6   Attack sensitivity to buffer capacity to limit

The smaller the expert buffer capacity limit, the easier it will be to find a set of adversarial tokens, $\tilde{X}^{\mathcal{A}}$, that block the assignment of tokens from $x^*$ to its preferred expert(s). We repeat our attack under different expert buffer capacity limits and present results in the Table 2.

Table 2: *Attack success for different expert buffer capacity limits.*

| Buffer capacity, $B_e$/Number of tokens in batch, $B \cdot T$ | Capacity variable, $C$ | Did the attack induce an error? |
| --- | --- | --- |
| 0.2 | 0.8 | Yes |
| 0.4 | 1.6 | Yes |
| 0.5 | 2 | No |
| 0.6 | 2.4 | No |
| 0.8 | 3.2 | No |
| 1 | 4 | No |

As expected, the attack is successful for smaller buffer capacity limits, and as the size of the buffer grows it becomes harder to affect expert assignments of the target $x^*$, causing the attack to fail.

# 7   Example of a denial-of-expert attack

Until now the attack goal has been to induce model output errors for a target input $x^*$. Overflowing an expert's buffer or changing the routing assignment (for deeper MoE layers in model) to cause such an error is a byproduct of this goal. We now show we can explicitly optimize to fill a chosen expert's buffer. We set the buffer capacity limit to 20 tokens per expert and optimize the adversarial data-points $X^{\mathcal{A}}$ to fill the buffer of the third expert (out of eight) in the first MoE layer by changing the objective function in Algorithm 2 to count how many tokens from $x^*$ are processed by this expert. Note, this changes the attack assumptions, as the adversary now needs white-box access to the model during the construction of $\tilde{X}^{\mathcal{A}}$ because they need to monitor an expert's buffer. From Figure 3, we see that initially 6 out of the 12 tokens from $x^*$ are processed by this expert; within 28 iterations we can find a batch of data-points $\tilde{X}^{\mathcal{A}}$ that cause all of these 6 tokens from $x^*$ to be dropped.

# 8   Mitigations

The attack relies on two optimizations made by MoE: (1) the usage of expert buffer capacity limits, and (2) batch dependent expert routing assignments. If the buffer capacity limit is removed, or set

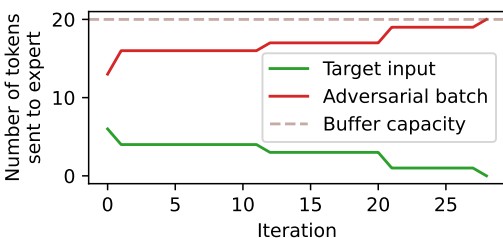

Figure 3: *Attack that constructs adversarial inputs that block the preferred expert of the majority of tokens from the target $x^*$.*

to the total number of tokens in the batch, then the attack will fail. We confirmed this by checking that our attack fails on the default routing strategy in the Mixtral-$8\times7B$ that does not set a buffer capacity limit. However, the use of expert buffer capacity limits is useful from a hardware utilization perspective as it avoids wasted computation and memory. We therefore list other mitigation methods, that will not offer guarantees of protection, but do decrease the efficiency of the attack. For each suggestion that follows, we confirmed that our random search attack fails when this mitigation is deployed.

**Randomize batch order.** The attack relies on the adversary controlling which position the target prompt $x^*$ appears in the batch. Model owners should always randomize the order of inputs in a batch. For particular sensitive queries where an increase in inference cost is tolerable, one could also run inference twice with two different orderings in the batch. If the two outputs on $x^*$ are significantly different, this should be a signal to the model owner that an inspection of the underlying causes is necessary.

**Use a large capacity slack, $C$.** As we have already seen from Table 2, using a larger capacity slack $C$ nullifies the attack as it becomes more difficult to fill an experts buffer with only tokens from adversarial inputs.

**Sampling from gate weights instead of selecting the top-$k$.** The expert assignments for a token $z$ are deterministic; the top-$k$ values of $g(z) = \{g_1(z), g_2(z), \ldots, g_n(z)\}$ are routed to their chosen experts. If the attack can infer which of the $k$ experts are selected for $z$, they can construct adversarial inputs to block those experts. Given that $g(z)$ outputs a probability distribution over $n$ experts, we could sample from this distribution to select $k$ experts rather than selecting the top-$k$. Sampling from $g(z)$ results in randomness in the expert assignments for $z$, making it harder for an attack to decide which expert buffers to fill.

## 9   Discussion

Combining queries from untrusted sources improves hardware utilization, but opens the door for exploitation. The machine learning community is well aware that dropping tokens due to buffer capacity limits, or routing tokens to non-preferred experts, can reduce performance (Fedus et al., 2022). We encourage the machine learning community to consider risks that come with inter-batch dependent outputs beyond subpar model performance.

It is unclear if an attack on MoE could represent a practical risk to deployed models. Although the practicality of our attack is limited in its current form, there are obvious directions for future work. Firstly, we expect that random search is extremely inefficient, and developing stronger gradient-based attacks could help model owners understand worst-case behavior. Secondly, another route to measuring worst-case behaviour is to record the variance in model outputs when tokens are dropped at each MoE layer, or tokens are assigned to random experts. If model outputs are robust to changes in expert routing then the risk of an attack diminishes. Thirdly, there are various routing strategies beyond the vanilla strategy we experiment with; it is important to understands the trade-offs between performance and security that come with these different routing choices. Fourthly, models that use MoE are commonly trained with load balancing auxiliary losses. This may help increase the robustness to an attack, as each expert may perform well at processing tokens across a broad range of topics. Future work could investigate how load balancing during training can help mitigate the

attack. Finally, we have not experimented with instruction-tuned models, which we expect to be more likely to be deployed in security sensitive settings (e.g. coding assistants).

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
