# OpenReview forum: "Buffer Overflow in Mixture of Experts"
_NeurIPS.cc/2024/Workshop/SafeGenAi — SafeGenAi Poster_

### Official Review · Reviewer_N13c · 2024-10-09
**Identifies critical MoE model vulnerabilities with clear demonstrations and mitigation strategies;**

**Rating:** 7
**Confidence:** 4

**Review:**

The paper "Buffer Overflow in Mixture of Experts" highlights a significant security vulnerability in Mixture of Experts (MoE) models, where expert routing strategies that incorporate cross-batch data can be exploited through adversarial queries to manipulate model outputs. The authors convincingly demonstrate this risk through a well-structured proof-of-concept attack in a toy setting and propose several mitigation strategies. While the experiments are somewhat limited to simplistic scenarios, the paper makes a substantial contribution to the field of machine learning security by addressing a previously underexplored issue. This study is valuable for its practical implications, though it could benefit from a broader experimental validation and a more detailed exploration of the mitigation techniques' trade-offs.